# A Phase 2, Single-Arm, Open-Label Clinical Trial on Adjuvant Peptide-Based Vaccination in Dogs with Aggressive Hemangiosarcoma Undergoing Surgery and Chemotherapy

**DOI:** 10.3390/cancers15174209

**Published:** 2023-08-22

**Authors:** Laura Marconato, Luca Tiraboschi, Marina Aralla, Silvia Sabattini, Alessia Melacarne, Chiara Agnoli, Andrea Balboni, Marta Salvi, Armando Foglia, Sofia Punzi, Noemi Romagnoli, Maria Rescigno

**Affiliations:** 1Department of Veterinary Medical Sciences, Alma Mater Studiorum University of Bologna, Ozzano dell’Emilia, 40064 Bologna, Italy; silvia.sabattini@unibo.it (S.S.); chiara.agnoli2@unibo.it (C.A.); a.balboni@unibo.it (A.B.); armando.foglia2@unibo.it (A.F.); sofia.punzi@studio.unibo.it (S.P.); noemi.romagnoli@unibo.it (N.R.); 2IRCCS Humanitas Research Hospital, 20089 Rozzano, Italy; luca.tiraboschi@humanitasresearch.it (L.T.); alessia.melacarne@humanitasresearch.it (A.M.); marta.salvi@humanitasresearch.it (M.S.); maria.rescigno@hunimed.eu (M.R.); 3Pronto Soccorso Veterinario Laudense, 26900 Lodi, Italy; marina.aralla@gmail.com; 4Department of Biomedical Sciences, Humanitas University, 20072 Pieve Emanuele, Italy

**Keywords:** translational research, angiosarcoma, canine, immunotherapy

## Abstract

**Simple Summary:**

Canine hemangiosarcoma shares many clinical, histologic and molecular features with human angiosarcoma. Thus, the dog represents a powerful and alternative model of spontaneously occurring hemangiosarcoma for comparative studies and developmental therapeutic investigation. In both species, surgery and adjuvant dose-intense chemotherapy represent the gold standard treatment. Several attempts have been made to improve patient outcomes. Novel clinical protocols have been designed, and different pharmacological and surgical approaches have been tested. However, hemangiosarcoma treatment remains an unmet medical need. Immunotherapy has emerged as a promising strategy. An anticancer vaccine was administered to 28 dogs with biologically aggressive hemangiosarcoma in combination with the standard treatment. The endpoints of the study were efficacy and safety. In addition to the tolerability of the vaccine, the induction of immune responses was reported, ultimately leading to improved outcomes compared with historical controls receiving the standard of care treatment only. The current findings provide promising potential for future management in both species.

**Abstract:**

To test the antitumor effect and safety of peptide-based anticancer vaccination in dogs with hemangiosarcoma undergoing the standard of care (SOC; surgery and doxorubicin), canine hemangiosarcoma cells were infected with *Salmonella typhi* Ty21a to release immunogenic endoplasmic reticulum stress-related peptides into the extracellular milieu via CX43 hemichannels opening. The infected tumor cell secretome constituted the vaccine. Following the SOC, dogs with biologically aggressive hemangiosarcoma were vaccinated a total of five times, once every 3 weeks, and were followed up with serial imaging. A retrospective population of dogs undergoing the SOC alone served as controls. The primary endpoints were the time to progression (TTP) and overall survival (OS), and the secondary endpoints were toxicity and immune responses. A total of 28 dogs were vaccinated along with the SOC, and 32 received only the SOC. A tumor-specific humoral response along with a vaccine-specific T-cell response was observed. Toxicity did not occur. The TTP and OS were significantly longer in vaccinated versus unvaccinated dogs (TTP: 195 vs. 160 days, respectively; *p* = 0.001; OS: 276 vs. 175 days, respectively; *p* = 0.002). One-year survival rates were 35.7% and 6.3% for vaccinated and unvaccinated dogs, respectively. In dogs with hemangiosarcoma undergoing the SOC, the addition of a peptide-based vaccine increased the TTP and OS, while maintaining a safe profile. Moreover, vaccinated dogs developed a tumor-specific response, supporting the feasibility of future phase three studies.

## 1. Introduction

Canine hemangiosarcoma is a malignant and highly metastatic tumor that originates from endothelial cells in various organs; more often, it arises in the spleen, the right atrium or auricle of the heart, skin and bone [1,2]. Given the non-specific symptoms, the diagnosis of hemangiosarcoma is challenging, often leading to identification at an advanced or metastatic stage of the disease [1]. However, in many cases, the diagnosis follows tumor rupture with hemoperitoneum or cardiac tamponade, depending on the primary site, which can result in death and necessitates urgent management [1]. The prognosis is generally poor and strongly influenced by the anatomic site, treatment and clinical stage at admission. Treatment typically requires a multimodal approach, combining surgery and systemic chemotherapy [1,2]. For dogs with splenic hemangiosarcoma treated with surgery alone, survival times are typically less than 2–3 months [3,4]. The administration of adjuvant chemotherapy aims to increase progression-free survival, and the existing literature indicates better survival for dogs undergoing a combinatorial approach, with reported median survival times of 5–8 months for dogs with non-metastatic hemangiosarcoma [1,4,5,6]. Due to the unfavorable anatomic location, cardiac hemangiosarcomas are almost always inoperable. If surgery is possible, the addition of adjuvant doxorubicin can extend survival to 6 months [7].

Canine hemangiosarcoma shares many clinical, histologic and molecular features with human angiosarcoma, which is equally aggressive and associated with limited survival [8,9,10]. One striking difference between the species is the incidence rate. Hemangiosarcoma is frequently diagnosed in dogs, but it is extremely rare in human beings, with only 200 cases of splenic angiosarcoma reported worldwide [11,12]. To date, there are no recommended standardized treatments or guidelines, and patients have historically been treated with surgery and/or chemotherapy.

Recently, immunotherapy has emerged as a valid combinatorial approach for various types of human cancer. Soft tissue sarcomas have traditionally been considered immunologically “cold” tumors. However, current studies are re-evaluating the immunotherapy potential in this field [13,14,15].

Immunotherapy has become a significant option in veterinary oncology as well [16,17,18,19]. Initial studies focusing on immunotherapy have suggested that canine hemangiosarcoma is immunogenic [20,21,22], paving the way to new treatment strategies. Previously, we showed that the *Salmonella* Tiphy (Ty21a) infection of human, murine and canine cancer cells promotes the release of non-conventional ER-stress-response-derived immunogenic peptides (ERstrePs) through connexin 43 (CX43) hemichannels. These peptides, if used in a vaccine formulation, are capable of eliciting an antitumor response in these model systems [23]. Further, a clinical study by our group documented that a heterologous, anticancer vaccine generated from primary osteosarcoma cells induced a tumor-specific immune response in dogs with appendicular osteosarcoma, significantly prolonging survival compared to dogs receiving the standard of care (SOC) treatment only [24].

The aim of this trial was to test the antitumor effect and safety of multiple dermal administrations of a peptide-based anticancer vaccine in dogs with biologically aggressive hemangiosarcoma undergoing SOC.

## 2. Materials and Methods

The Ethical Committee of the University Veterinary Hospital (Bologna, Italy) approved this study (protocol 7638). Written consent for participation in the trial was obtained from all dog owners. The vaccine was provided free of charge, and dog owners were responsible for covering all other medical expenses.

### 2.1. Trial Design

This was a bi-center, single-arm, open-label prospective trial of a peptide-based anticancer vaccine in dogs with newly diagnosed, biologically aggressive hemangiosarcoma presented to the University of Bologna and Pronto Soccorso Veterinario Laudense.

Client-owned dogs of any age, sex, weight or breed with surgically resected, biologically aggressive hemangiosarcoma (including splenic, cardiac, osseous and retroperitoneal) were considered for inclusion.

To be eligible for recruitment, dogs had to be metastasis-free at the beginning of chemotherapy. Staging work-up consisted of history and physical examinations, complete differential blood cell counts, serum biochemistry profiles, coagulation profiles, urinalysis, thoracic radiographs and abdominal ultrasound or total body CT (TBCT), and echocardiography.

After surgical removal of the primary hemangiosarcoma, dogs had to be treated with doxorubicin (administered at 30 mg/m^2^ IV for dogs weighing >15 kg and 1 mg/kg for those weighing ≤15 kg) every 3 weeks for 4 cycles. Twenty-four hours after the second doxorubicin administration, dogs were allowed to start immunotherapy.

Dogs were not eligible for enrollment if they had previously undergone chemotherapy, if they had clinically relevant comorbidities that would limit their life expectancy or if they had a dermal or lingual hemangiosarcoma due to their more benign biological behavior [25,26]. Treatment with nonsteroidal anti-inflammatory drugs or analgesics was allowed before enrollment and during the trial. Permission to perform a necropsy at the time of death from any cause was requested.

### 2.2. Vaccine Preparation

Primary canine hemangiosarcoma cells were obtained from the dissociation of canine tumor tissues; surgical specimens were obtained from hemangiosarcoma occurring in the spleen, omentum, liver, intestine, bone and lung. Tissues were minced with a scalpel in a cell strainer, and the derived single-cell suspension was washed and filtered with a cell strainer (100 µm). If needed, red blood cell lysis was performed using an adequate buffer (Red Blood Cell Lysis Solution, Friedrich-Ebert-Straße 68 51429 Bergisch Gladbach, Germany, cat. 130-094-183). Cells were then counted and cultured at a high concentration in Dulbecco’s Modified Eagle Medium (DMEM) supplemented with a United States (U.S.) origin 10% Fetal Bovine Serum (FBS), 2 mM glutamine, 1× penicillin-streptomycin solution (complete medium). Vivofit^®^ (Thyphoid vaccine live oral Ty21a) is a commercial vaccine containing the attenuated strain of *Salmonella* enterica serovar typhi Ty21a and is grown at 37 °C in Luria broth.

The vaccine was prepared as previously described [23]. Briefly, single bacterial colonies were grown overnight and restarted the next day to reach an absorbance at 600 nm of 0.6, corresponding to 0.6 × 109 colony-forming units (CFUs)/mL. Canine hemangiosarcoma cells (2 × 106 cells/mL) were incubated with bacteria for 90 min in tubes at a cell-to-bacteria ratio of 1:50, in the appropriate medium containing L-Glutamine without antibiotics. Cells were washed and incubated in a medium supplemented with gentamicin (50 mg/mL) in tubes for 18 h to kill extracellular bacteria.

During incubation, immunogenic peptides were released via CX43 hemichannels by cancer cells in the extracellular space; the supernatant enriched with immunogenic peptides (secretome) was collected and filtered through a 0.22 µm filter to eliminate remaining potentially live bacteria. Every dose of the vaccine was assembled by lyophilizing 1 mL of secretome.

### 2.3. Immunofluorescence

After the infection protocol, untreated and *Salmonella*-infected hemangiosarcoma primary cells were plated at a concentration of 100,000 cells/mL on Poly-D-Lysine-treated polarized microscope slides (kept in separated sterile plates to avoid contamination). After 4 h, adherent cells were washed with PBS and fixed in 4% PFA for 20 min. Fixed cells were then rehydrated and blocked with 0.1 M Tris-HCl at a pH of 7.4 with 2% FBS and 0.3% Triton X-100. Subsequently, cells were stained using mouse anti-CX43 (1:100, Invitrogen, 138,300, Waltham, MA, USA) and goat anti-Lipopolysaccharide (1:100, Antibodies-online, ABIN479062) antibodies overnight. After three washes in 0.1 M Tris-HCl at a pH of 7.4 with 0.05% Tween20, samples were stained with donkey anti-mouse-AF647 conjugates (Thermofisher, 3747 N. Meridian Road Rockford, IL, USA, A21202), donkey anti-goat-AF488-conjugaded antibodies (Thermofisher, A11055) and rabbit anti-alpha smooth muscle Actine-AF555-conjugated antibodies (Abcam, ABIN479062). Cell nuclei were counterstained with DAPI (1:45,000, Thermofisher, 1306). Sections were mounted using the Vectashield mounting medium and were acquired with the Leica TCS SP8 Laser scanning confocal microscope HC PL APO CS2 40×/1.30 oil immersion objective (Leica). Representative images and CX43 quantifications were performed with Fiji software 2.9.0/1.53t (ImageJ).

### 2.4. ATP Release

CX43-hemichannels opening upon *Salmonella* infection was assessed by measuring the ATP released in extracellular milieu through the Cell-Titer-Glo Luminescent cell viability assay (Promega, G7572). In brief, the supernatants of untreated and *Salmonella*-infected cells were distributed in 96-well white clear-bottom plates (Corning, 3610). Subsequently, supernatants were mixed with the MIX-assay solution in an equivalent volume. Finally, luminescence emission was assessed using a luminometer.

### 2.5. Vaccine Administration

Twenty-four hours following the second doxorubicin administration, dogs were vaccinated. In the absence of disease progression, a minimum number of five dermal injections, one every 3 weeks, was scheduled for each enrolled dog. For the first and second immunizations, the lyophilized vaccine was dissolved in Nobivac Lepto (MSD Animal Health—used as an adjuvant), whereas the remaining three doses were diluted in sterile water.

Before each vaccination, imiquimod cream (Aldara 5% cream—a TLR7/8 agonist) was rubbed topically at the site of the vaccination on a 3 × 3 cm shaved and clean area as an additional adjuvant. After 15 min, the vaccine (1 mL in total volume) was administered intradermally in close proximity to a peripheral lymph node.

Only 1 dog received an autologous vaccine (derived from its own tumor cells), and 27 dogs received a heterologous vaccine (which was prepared by using the hemangiosarcoma cells of another dog). The dog that received the autologous vaccine had a primary osseous hemangiosarcoma, and it was the first case from which the hemangiosarcoma tumor line was isolated. Subsequent vaccine doses were obtained from this case.

### 2.6. Immunomonitoring

For immunomonitoring, the vaccine-specific T-cell response (from patient-derived Peripheral Blood Mononuclear Cells, PBMCs) and the tumor-specific humoral response (from the patient-derived serum) were tested. An amount of 10 mL of whole blood and 5 mL of serum were collected longitudinally at four timepoints across the five vaccinations, namely before the first, second, third and fifth vaccine administrations. Immediately after withdrawal, the blood samples were processed for PBMC isolation and then stored in a liquid nitrogen tank. The serum was stored at −80 °C.

### 2.7. Vaccine-Specific T-Cell Response

PBMCs collected before and at the last vaccination were tested for their specific activation against the vaccine. In particular, we assessed their ability to release IFNγ upon vaccine stimulation. Stored PBMCs were thawed, counted and plated into flat-bottom plates in Roswell Park Memorial Institute (RPMI) medium supplemented with a 10% FBS, 2 mM glutamine, 1× penicillin-streptomycin solution (complete medium). Upon thawing, the culture medium was added with 1 μg/mL of ConA (Sigma, C2010, St. Louis, MO, USA) as a survival stimulus; every other day, IL-2 (100 U/mL, Proleukin, Clinigen Healthcare B.V., Schiphol, The Netherlands) was added to sustain T-cell viability. After 7–10 days, at the end of ConA stimulation, cells were washed twice and plated with 2 × 105 cells/well, with or without the vaccine (the same batch used to immunize dogs) in a 96-well plate. IFN-γ released by T-cells was measured after 72 h of incubation via ELISA (canine IFN-γ, R&D, CAIF00).

### 2.8. Humoral Response

Tumor cell lysate was obtained from the same primary hemangiosarcoma cells used to generate the vaccine. Briefly, cells resuspended in PBS at a known concentration underwent lysis via the freeze and thaw protocol. Each well of the ELISA plate was coated overnight at 4 °C by the lysate generated from 8 × 10^4^ cells. Plates were then incubated for 2 h at room temperature in a 5% BSA solution to block non-specific interactions. Then, several dilutions of each serum were incubated on top of the cell lysate for 2 h, and the humoral response was assessed with the HRP-conjugated anti-canine IgG antibody (Jackson ImmunoResearch, West Grove, PA, USA).

### 2.9. Control Group (SOC)

Dogs treated at the University of Bologna before the start of the trial that underwent the SOC, consisting of surgery and 4 to 6 cycles of adjuvant doxorubicin at the same dosage (i.e., 30 mg/m^2^ IV for dogs weighing >15 kg and 1 mg/kg for those weighing ≤15 kg every 3 weeks), and that were metastasis-free at the beginning of chemotherapy, served as retrospective study cohort controls [27]. Metastasis was ruled out by means of thoracic radiographs, abdominal ultrasound, echocardiography, TBCT and/or exploratory surgery.

Dogs were selected starting from January 2019 to allow for a period of up to a 1-year clinical follow-up for any given case.

### 2.10. Antitumor Response Assessment and Follow-Up

For vaccinated dogs, disease status was ascertained via imaging using standard techniques at each vaccination and on a monthly–bimonthly basis thereafter. For control dogs, disease status was ascertained via imaging after every two cycles of chemotherapy, at the end of treatment and after every 2–3 months thereafter until the documentation of progressive disease. When clinically indicated, additional diagnostics were performed.

### 2.11. Toxicity

Adverse events related to chemotherapy and vaccination were graded according to VCOG [28].

### 2.12. Sample Size Calculation

In the trial design phase, a power analysis was performed to estimate the required number of subjects to detect a difference in the time to progression (TTP) of 3 months (SOC: 150 days, SOC + VAX: 240 days) using a two-sided log-rank test at 80% power with a *p*-value ≤ 0.05. A sample size of 20 cases and 20 controls was deemed adequate. Based on tolerability and observed clinical benefit, case accrual ultimately continued to 28 dogs.

### 2.13. Statistical Analysis

For descriptive statistics, data were tested for normality with the D’Agostino and Pearson omnibus normality test. Normally distributed variables were expressed as the mean ± SD, whereas non-normally distributed variables were expressed as the median with a range.

Recorded information included signalment (i.e., breed, age, sex and weight), tumor location and clinical stage at admission.

The distribution of demographic data and possible prognostic variables in the cases (SOC + VAX) and controls (SOC) were compared with the Mann–Whitney U test or the chi-squared test/Fisher’s exact test for continuous and categorical variables, respectively.

The TTP was computed from the date of surgery to the occurrence of local tumor progression or metastasis or to the last visit. Overall survival (OS) was calculated from the date of surgery to the date of death for any cause or to the last visit. If progression or death did not occur, cases were censored.

Survival plots were generated according to the Kaplan–Meier product limit method. The TTP and OS were compared between the two groups with the log-rank test. Outcome was also evaluated after stratification based on the most represented anatomic locations (spleen and heart).

The 6-month and 1-year survival rates were calculated for each group.

The analysis was carried out with the use of commercial statistical software (SPSS Statistics v. 26, IBM, Somers, NY, USA, and Prism v. 5.0, GraphPad, San Diego, CA, USA). The significance level was set at *p* ≤ 0.05.

## 3. Results

### 3.1. Salmonella Infection Leads to Gap Junction Hemichannel Opening in Hemangiosarcoma Cells

A fundamental feature of obtaining peptide release in the secretome is the upregulation of CX43 and the opening of gap junction hemichannels (23). In agreement with a previous study on osteosarcoma (24), the *Salmonella* infection of hemangiosarcoma cells prompted CX43 hemichannel overexpression (Figure 1A,B). To address the functional opening of hemichannels, we evaluated ATP release in the culture supernatant. As shown in Figure 1C, ATP was promptly released upon *Salmonella* infection, thereby confirming hemichannel opening. This release was not due to cell death, as toxicity was not detected after *Salmonella* treatment (Figure 1D).

### 3.2. Vaccinated Dogs (SOC + VAX)

From October 2020 to July 2022, 28 dogs were enrolled. There were 16 splenic hemangiosarcomas (12 ruptured), 7 cardiac hemangiosarcomas (all with auricle tumor), 4 retroperitoneal hemangiosarcomas and 1 osseous hemangiosarcoma.

The demographic information and tumor locations of the enrolled dogs are provided in Appendix A.

All dogs underwent surgery and received four doses of doxorubicin at a dosage of 30 mg/m^2^ IV every three weeks. All dogs were confirmed to be free of metastases at the time of the first chemotherapy administration.

Chemotherapy-related adverse events included five episodes of grade 1 gastrointestinal toxicity, one episode of grade 2 gastrointestinal toxicity, one episode of grade 3 neutropenia and one episode of hypersensitivity.

### 3.3. Vaccine Administration

The median number of vaccines received was 5 (range, 2 to 11). Overall, 22 dogs received all five planned vaccinations. Among the remaining six dogs, two received four vaccinations, two received three vaccinations, and two received two vaccinations. In all of them, the reason for not completing the vaccination protocol was disease progression and tumor-related death.

Thirteen dogs that were metastasis-free after five vaccine doses were allowed to receive prolonged vaccination. They continued intradermal vaccination monthly and received 6–11 total vaccinations. In total, 175 vaccine doses were administered without any documented toxicity.

### 3.4. Immunomonitoring

The humoral response was evaluated in all dogs by comparing tumor-specific IgG levels at baseline and after the fifth vaccination (Figure 2A). On the other hand, T-cell-specific activation upon stimulation with the vaccine was analyzed in 18 dogs with viable PBMCs after the fifth vaccination (Figure 2B). However, in 10 cases, PBMCs were not sufficiently viable, leading to the exclusion of those patients from the analysis. No correlation between immune response and the TTP was observed. However, we observed a clear induction of both humoral and T-cell vaccines and tumor-specific responses, indicating that the vaccine served its purpose.

### 3.5. Retrospective Control Group (SOC)

A control group of 32 dogs that had undergone surgery and chemotherapy were retrospectively evaluated (Appendix A). There were 22 splenic hemangiosarcomas (17 ruptured), 5 cardiac hemangiosarcomas (all with auricle tumor), 3 retroperitoneal hemangiosarcomas and 2 osseous hemangiosarcomas. None had developed metastatic disease at the beginning of chemotherapy.

Chemotherapy-related adverse events were recorded in eight (25%) dogs and included three episodes of grade 1 gastrointestinal toxicity, two episodes of grade 1 neutropenia, two episodes of grade 2 gastrointestinal toxicity and one episode of grade 3 gastrointestinal toxicity.

No significant differences were observed with regard to demographics and prognostic variables between vaccinated dogs and the control group (Table 1).

### 3.6. Oncologic Outcome

Among vaccinated dogs, 4 (14.3%) had developed metastatic disease before vaccination, and 17 (60.7%) additional dogs developed metastatic disease in the liver (n = 8), lungs (n = 3), muscles and spleen (n = 2), muscles and lungs (n = 1), liver and pancreas (n = 1), omentum (n = 1) and heart (n = 1). One dog with cardiac hemangiosarcoma and one dog with bone hemangiosarcoma experienced local recurrence without any evidence of distant metastases.

In the control group, 30 (93.8%) dogs developed distant metastases in the liver (n = 12), peritoneum (n = 10), lung (n = 4), heart (n = 1), brain (n = 1), humerus and spleen (n = 1) and kidney and lungs (n = 1). Two dogs with cardiac hemangiosarcoma (n = 1) and retroperitoneal hemangiosarcoma (n = 1) experienced local recurrence.

The overall median TTP was 195 days (95% CI, 139–251) in the SOC + VAX group and 160 days (95% CI, 127–193) in the SOC group (*p* = 0.001; Figure 3; Table 1).

Considering splenic hemangiosarcoma, the median TTP was 266 days (95% CI, 76–456) in the SOC + VAX group and 156 days (95% CI, 100–212) in the SOC group (*p* = 0.005; Figure 4; Table 1).

For cardiac hemangiosarcoma, the median TTP was 181 days (95% CI, 171–191) in the SOC + VAX group and 60 days (95% CI, 36–84) in the SOC group (*p* = 0.011; Figure 5; Table 1).

Upon the closure of data analysis, 4 (14.3%) SOC + VAX dogs were alive after a median follow-up of 668 days (range, 552–847), and 24 (85.7%) had died. The cause of death was attributable to hemangiosarcoma in 23 dogs (95.8%) and to tumor-unrelated causes in 1 dog (4.2%). Necropsy excluded hemangiosarcoma in this dog.

The four dogs with developing metastatic diseases before vaccination survived 125, 160, 306 and 394 days (median, 233 days). Among them, one dog experienced a pulmonary complete response, whereas the remaining three were stable.

Thirty-one (96.9%) SOC dogs died due to hemangiosarcoma metastasis and/or local recurrence.

The median OS was 276 days (95% CI, 190–362) in the vaccine group and 175 days (95% CI, 128–222) in the control group (*p* = 0.002; Figure 3; Table 1).

Considering splenic hemangiosarcoma, the median OS was 269 days (95% CI, 140–398) in the SOC + VAX group and 190 days (95% CI, 144–236) in the SOC group (*p* = 0.004; Figure 4; Table 1). For cardiac hemangiosarcoma, median OS was 284 days (95% CI, 71–497) in the SOC + VAX group and 100 days (95% CI, 0–255) in the SOC group (*p* = 0.011; Figure 5; Table 1).

The overall 6-month survival rates were 71.4% for SOC + VAX dogs and 50% for SOC dogs. The 1-year survival rates were 35.7% and 6.3%, respectively. When only splenic hemangiosarcoma was considered, the 6-month survival rates were 68.8% for SOC + VAX dogs and 54.5% for SOC dogs. The 1-year survival rates were 43.8% and 4.6%, respectively. When only cardiac hemangiosarcoma was considered, the 6-month survival rates were 71.4% for SOC + VAX dogs and 0.0% for SOC dogs. The 1-year survival rates were 28.6% and 0.0%, respectively.

## 4. Discussion

The optimal management of dogs with biologically aggressive hemangiosarcoma presents a challenging clinical dilemma, with no effective therapeutic options available [1,2]. Based on our findings, immunotherapy may offer a new complementary approach as a first-line therapy for treating hemangiosarcoma.

This study was designed and powered to assess the primary endpoints of the TTP and OS, and it successfully identified a significant difference between the two treatment groups: SOC versus SOC + VAX. The peptide-based heterologous vaccine, when combined with SOC, significantly extended the TTP (195 days) and OS (284 days) compared to dogs receiving the SOC only (160 and 175 days, respectively).

Four (14.3%) vaccinated dogs showed no radiographic and clinical progression after a median of 668 days (range, 552–847). All of these dogs remained on trial for more than 18 months without progression, meeting the trial endpoint that warrants further investigation.

The overall 6-month and 1-year survival rates of 71% and 36%, respectively, indicate a stable treatment response and a promising long-term effect. When specifically looking at splenic hemangiosarcoma, the 6-month survival and 1-year survival rates were 69% and 44%, respectively. Although this represents a small subset, these outcomes compare favorably with our control population and previously published data, which put 1-year survival at rates below 10% [29,30].

Four dogs in the SOC + VAX group developed metastatic disease before vaccination. It has been shown that dogs with measurable disease receiving the SOC have a dismal prognosis, with a median survival time of 140 days [31]. In the current series, vaccinated dogs with metastatic disease had a median survival time of 233 days, suggesting that vaccination leads to survival prolongation even in the advanced-stage population. Further subgroup analysis related to the primary endpoint indicated that splenic hemangiosarcoma possibly shows a greater survival advantage compared with cardiac hemangiosarcoma. This difference may be related to a lower likelihood of achieving complete tumor removal in the cardiac forms, resulting in a higher tumor burden remaining behind. In fact, for auricle hemangiosarcoma, radical surgery is never possible, and the margins are inevitably infiltrated.

The vaccine was well tolerated, and no adverse events were reported. Moreover, the vaccine demonstrated immunogenicity. After completing the vaccine schedule, the dogs developed a vaccine-specific immune response. We confirmed the induction of both T-cell and B-cell responses by evaluating the dogs’ PBMCs and serum, respectively. No correlation between immune-response and the TTP was observed, possibly due to the heterogenous site of tumor onset.

To induce an effective immune response against cancer antigens, it is necessary to overcome immunological tolerance by combining immunogenic antigens with a potent adjuvant [30]. Our vaccine is obtained from the secretome of *Salmonella*-infected primary tumor cells, which is enriched with ERstrePs. This class of peptide consists of shared non-mutated tumor antigens expressed only by ER-stressed tumor cells, but not by healthy (non-ER-stressed) cells, and that can boost a robust anti-tumor response while sparing healthy cells from being targeted and killed [23].

The clinical response observed in all vaccinated dogs, including the four animals with measurable metastatic diseases at the beginning of immunotherapy, indicates that the vaccination induced an adaptive immune response, promoting long-term acquired immunity.

To downstage the disease, dogs underwent surgery followed by doxorubicin. Unlike other chemotherapeutic drugs, doxorubicin possesses immunomodulatory properties, enabling it to regulate the content of T-lymphocytes and stimulate antigen-presenting dendritic cell function. As a result, it promotes antitumor immunity and enhances the response to immunotherapy [32,33,34,35,36].

In the current study, the vaccine was administered 24 h after the second doxorubicin dose. The first and second doses of the vaccine were dissolved in Nobivac Lepto, which served as an adjuvant to enhance the CD^4+^ immune response. Dogs are routinely vaccinated against Leptospira; thus, we exploited the immune activation against Leptospira antigens to boost the priming of peptides within the anticancer vaccine [37,38]. Additionally, before each vaccination, imiquimod, a toll-like receptor 7/8 agonist, was topically applied to recruit and activate immune cells at the vaccination site, following the approach described in human immunotherapy trials [39].

Peptide-based “off-the-shelf” cancer vaccines offer many advantages, including high specificity, low manufacturing costs and a proven safety profile, making this strategy user-friendly, practical and affordable. The main challenge is the incorporation of multiple epitopes recognized by CD^4+^ and CD^8+^ cells, considering tumor heterogeneity. Here, we addressed this challenge by using the entire secretome of tumor cells and the unique characteristic of the released peptides associated with ER stress. ERStreP peptides specifically target tumor cells under significant stress, which often must cope with multiple mutations and gene duplications. This vaccine was specifically developed for canine use and tailored for hemangiosarcoma, making it particularly appealing and safe for this specific application.

The limitations of the study include the use of a retrospective population of dogs as controls and a modest sample size.

Moreover, the initial staging relied more on thoracic radiographs and ultrasound in unvaccinated dogs compared to vaccinated dogs, raising the possibility of missed metastatic diseases at the inclusion of dogs receiving the SOC.

Originally, the plan was for each dog to receive five total vaccinations. Nevertheless, on compassionate grounds, 13 dogs received additional doses, and no toxicity was reported. It remains unanswered if and how frequently a dog should be boosted.

Last, T-cell-specific activation upon stimulation with the vaccine was analyzed in 18 dogs only. Unfortunately, in 10 cases, PBMCs were not sufficiently viable, leading to the exclusion of those patients from the analysis.

Owing to the low disease incidence, human angiosarcoma is considered a rare cancer, thereby inevitably facing additional challenges, including a lack of clinical expertise and a lack of research interest. In consideration of the many similarities, the use of relevant animal models may overcome these challenges. The dog serves as a robust and alternative model of spontaneously occurring hemangiosarcoma for comparative studies and therapeutic investigation, as these tumors develop spontaneously in a host that is immunologically outbred and shares environmental exposure with humans [40,41,42]. Whether this pre-clinical vaccination strategy can be effectively translated to humans with angiosarcoma is currently unknown; nevertheless, the encouraging outcomes obtained in the canine species justify further consideration of vaccine therapy in the management of the disease. Indeed, translation to human patients is currently being pursued in our laboratories.

## 5. Conclusions

In dogs with biologically aggressive hemangiosarcoma undergoing the SOC, the addition of a bacteria-based vaccination strategy effectively controlled metastatic development, leading to prolonged survival while maintaining a safe profile. Vaccinated dogs also developed a sustained immune response, as documented by serial immunomonitoring. These results offer promising potential for future management, contributing to the design of larger and more definitive studies of vaccine therapies in earlier stages of disease. The validation of this approach in canine hemangiosarcoma provides crucial new data regarding treatment that may also be of benefit for the human disease.

## Figures and Tables

**Figure 1 cancers-15-04209-f001:**
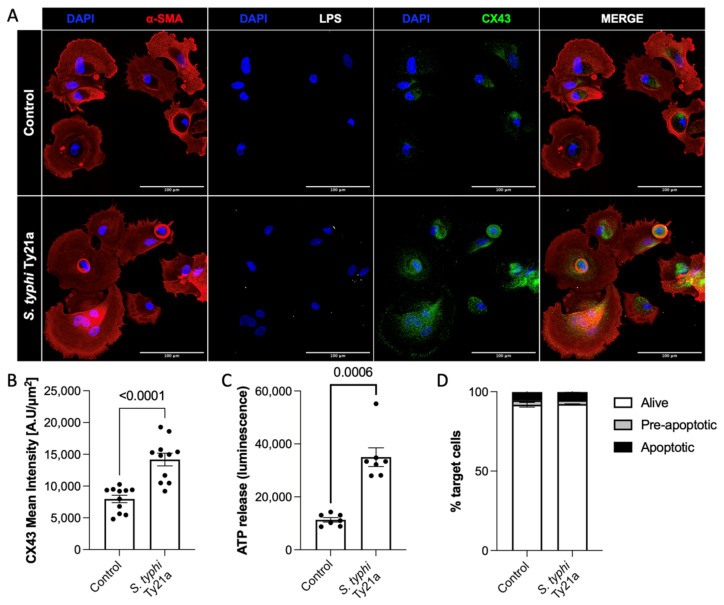
*S. typhi* Ty21a infection induces upregulation and opening of CX43 hemichannels on hemangiosarcoma primary cells, allowing the release of immunogenic peptides. Representative images of untreated and *Salmonella*-infected hemangiosarcoma cells (**A**) and quantification of CX43 expression in 4 independent experiments. Mann–Whitney, *p* < 0.0001 (**B**). Quantification of ATP released upon *Salmonella* infection. Mann–Whitney, *p* = 0.0006 (**C**). Annexin V-7AAD staining, showing hemangiosarcoma cell viability (**D**).

**Figure 2 cancers-15-04209-f002:**
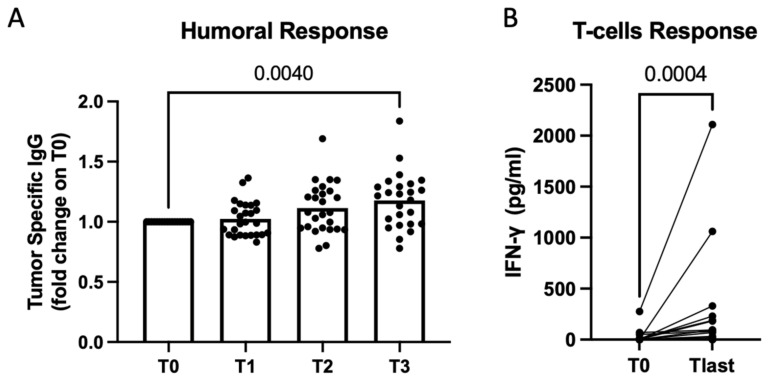
Secretome derived from *Salmonella*-infected hemangiosarcoma cells used as a vaccine to induce an efficient immune response in dogs. Immunomonitoring was performed on sera and PBMCs of vaccinated dogs at several timepoints (T0 before the first vaccination, T1 after 1 dose, T2 after 2 doses, T3 after 5 doses). Scatterplot showing quantification of tumor-specific IgG in dogs’ sera through ELISA. Data are shown as normalized at T0. Multiple comparisons mixed model, *p* = 0.004 (on graph). Vaccination resulted in statistical significance, also according the mixed-effects model, *p* = 0.0013 (**A**). Quantification of IFN-γ produced by dogs’ PBMCs upon vaccine stimulation. Raw data expressed as OD at 450 nm are shown for PBMCs collected before vaccination (T0) and at the latest timepoint (Tlast). Wilcoxon matched-pairs signed-rank test, *p* = 0.0004 (**B**).

**Figure 3 cancers-15-04209-f003:**
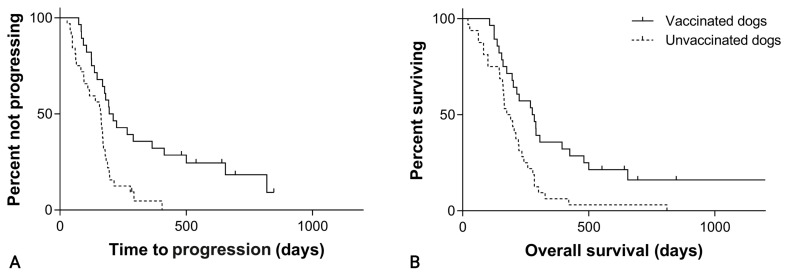
(**A**) Time to progression in 60 dogs with hemangiosarcoma treated with surgery and chemotherapy, grouped according to whether they received adjuvant peptide-based vaccination (solid line) or not (dashed line; *p* = 0.001). (**B**) Overall survival in 60 dogs with hemangiosarcoma treated with surgery and chemotherapy, grouped according to whether they received adjuvant peptide-based vaccination (solid line) or not (dashed line; *p* = 0.002).

**Figure 4 cancers-15-04209-f004:**
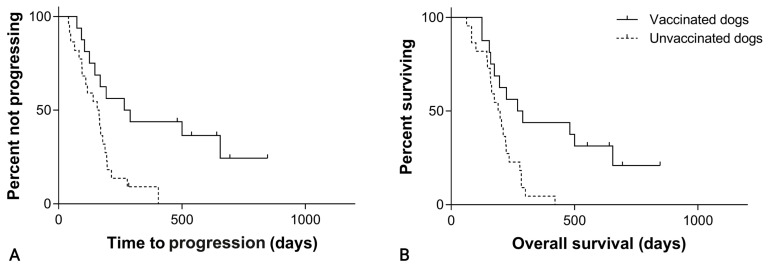
(**A**) Time to progression in 38 dogs with splenic hemangiosarcoma treated with splenectomy and chemotherapy, grouped according to whether they received adjuvant peptide-based vaccination (solid line) or not (dashed line; *p* = 0.005). (**B**) Overall survival in 38 dogs with splenic hemangiosarcoma treated with splenectomy and chemotherapy, grouped according to whether they received adjuvant peptide-based vaccination (solid line) or not (dashed line; *p* = 0.004).

**Figure 5 cancers-15-04209-f005:**
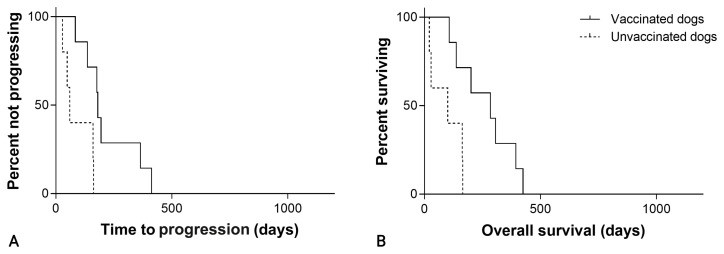
(**A**) Time to progression in 12 dogs with cardiac hemangiosarcoma treated with surgery and chemotherapy, grouped according to whether they received adjuvant peptide-based vaccination (solid line) or not (dashed line; *p* = 0.011). (**B**) Overall survival in 12 dogs with cardiac hemangiosarcoma treated with surgery and chemotherapy, grouped according to whether they received adjuvant peptide-based vaccination (solid line) or not (dashed line; *p* = 0.011).

**Table 1 cancers-15-04209-t001:** Baseline characteristics and survival analysis of 60 dogs with biologically aggressive hemangiosarcoma treated with surgery and chemotherapy with or without the addition of a peptide-based anticancer vaccine.

Variable	Vaccinated Dogs (n = 28)	Non-Vaccinated Dogs (n = 32)	*p*
Sex			0.212
Male	13 (46.4%)	20 (62.5%)	
Female	15 (53.6%)	12 (37.5%)	
Age (years)			
Median (range)	9 (5–12)	9.5 (3–15)	0.487
Weight (kg)			
Median (range)	27.6 (11.0–48.0)	29.8 (8.1–60.0)	0.892
Tumor location			0.6
Spleen	16 (57.1%)	22 (68.8%)	
Heart	7 (25.0%)	5 (15.6%)	
Other	5 (17.9%)	5 (15.6%)	
Ruptured splenic mass			>0.999
Yes	12 (75.0%)	17 (77.3%)	
No	4 (25.0%)	5 (22.7%)	
Treatment-related toxicity (grade ≥3)			>0.999
Yes	1 (3.6%)	1 (3.1%)	
No	27 (96.4%)	31 (96.9%)	

## Data Availability

The data that support the findings of this study are available from the corresponding author upon reasonable request.

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
