# Peer review of "A Phase 2, Single-Arm, Open-Label Clinical Trial on Adjuvant Peptide-Based Vaccination in Dogs with Aggressive Hemangiosarcoma Undergoing Surgery and Chemotherapy"

_cancers, 2023, doi:10.3390/cancers15174209_

Round 1

Reviewer 1 Report

This is an interesting and well-written manuscript evaluating immunotherapy as adjuvant in the treatment of canine hemangiosarcoma. Some small improvements can be made.

General comments

*) Please describe and define your control group consistently throughout the manuscript. The control group was evaluated as a retrospective study cohort and was neither a historical control, where you only cite published outcome data only, nor a second prospective treatment arm.

*) Why did you give about 50% of all vaccinated dogs more than the planned 5 vaccinations? All dogs had to be metastasis free and the primary tumor had to be surgically removed to enter the study? So how did you judge “benefit from the vaccine” in this phase of therapy? In a phase 2 trial it is really not ideal if you then claim to evaluate efficacy and safety – did you look at differences in TTP and OS between dogs receiving 5 vaccinations and those that received more?

*) You had 12 cardiac hemangiosarcomas – how was surgery performed in these cases? Was this a biopsy only without removing the tumor?

*) You had dogs below 15kg and even below 10kg – did these dogs all receive 30mg/m2 doxorubicin?

*) In the simple summary, abstract and throughout the manuscript a semicolon is used several times to separate sentences. Please consider using comma or point instead.

Keywords

The word hemangiosarcoma is already used in the title, so you can delete it as keyword

Simple summary

I suggest to replace “our” and “we” in the simple summary (L20, L22)

The definition “historical control” (L23) is misleading, see general comments

Abstract

Please define SOC already in L27 – you wait with a more precise information until L32 (Surgery and doxorubicin) in the abstract. Please define also in the abstract that the control group is a retrospective cohort and not a second prospective study arm.

Introduction

L72 – please reword “present-days studies”

L74 and L76: citation are not in square brackets?

Material and methods

L100 (p 3): please define “staged negative” or reword

L169 (p 4): Please add that imiquimod was administered at the site of the vaccination, this information is missing here and only mentioned in the discussion. Please give the volume that was administered intradermally.

L 186 (p 4): Please replace “IN” by “In”

Results

Figure 1: It seems the legend to 1D is missing?

L280 (p 7): the “end of the vaccination schedule” was then after 5 vaccinations or after the real end? (which in 13 dogs was later after 6-11 vaccinations)

Figure 2: Please explain in 2A T0, T1, etc (L288) as you do it for 2B (L293)

L295 Control group: I would consider to present the vaccinated group first and afterwards present the control group, as the control was not a second treatment arm of this study, but a retrospective cohort

Table 1: I would consider to remove the TTP and ST information in the table, as this information is at the moment repeated three times in the manuscript, in Table 1, in the main text and as Figure 3-5.

Fig 3-5: please reword “stratified”, as these is misleading; there were no two treatment arms and no true stratification

Discussion

L394: Please be consistent with abbreviation of survial time, I see ST, OS and TSS at the moment

some small suggestions are made in the reviewer comments

Author Response

This is an interesting and well-written manuscript evaluating immunotherapy as adjuvant in the treatment of canine hemangiosarcoma. Some small improvements can be made.

General comments

*) Please describe and define your control group consistently throughout the manuscript. The control group was evaluated as a retrospective study cohort and was neither a historical control, where you only cite published outcome data only, nor a second prospective treatment arm.

AUTHORS: we totally agree with this comment. The control group has been defined as “retrospective study cohort group” throughout the whole manuscript.

*) Why did you give about 50% of all vaccinated dogs more than the planned 5 vaccinations? All dogs had to be metastasis free and the primary tumor had to be surgically removed to enter the study? So how did you judge “benefit from the vaccine” in this phase of therapy? In a phase 2 trial it is really not ideal if you then claim to evaluate efficacy and safety – did you look at differences in TTP and OS between dogs receiving 5 vaccinations and those that received more?

AUTHORS: thank you for this relevant comment. As stated in the manuscript, “13 dogs were given prolonged vaccination on compassionate grounds, as there was a possibility that they had a benefit from the vaccine.” Hemangiosarcoma is a serious and life-threatening disease and represents an unmet medical need. Dogs that were metastasis-free at the end of the 5 doses were allowed to receive additional vaccination with the aim of reducing the chance of progression. While the vaccine is still in clinical development, and in the absence of other successful treatments, as well as with owners' compliance, we have decided to continue vaccinating dogs that have shown a good response. The following has been better clarified in the manuscript: “Thirteen dogs that were metastasis-free at the end of the 5 doses were allowed to receive additional vaccination.”

*) You had 12 cardiac hemangiosarcomas – how was surgery performed in these cases? Was this a biopsy only without removing the tumor?

AUTHORS: all dogs with cardiac hemangiosarcoma (in both the vaccine and control groups) had the tumor involving the auricle. The entire auricle was removed in these dogs. This has been clarified.

*) You had dogs below 15kg and even below 10kg – did these dogs all receive 30mg/m2 doxorubicin?

AUTHORS: thank you for this comment. You are perfectly right. Dogs weighing <15 kg received doxorubicin @ 1 mg/kg. This has been added.

*) In the simple summary, abstract and throughout the manuscript a semicolon is used several times to separate sentences. Please consider using comma or point instead.

AUTHORS: done.

Keywords

The word hemangiosarcoma is already used in the title, so you can delete it as keyword

AUTHORS: we changed “hemangiosarcoma” into “angiosarcoma” as keyword.

Simple summary

I suggest to replace “our” and “we” in the simple summary (L20, L22)

AUTHORS: done.

The definition “historical control” (L23) is misleading, see general comments

AUTHORS: agreed- this has been changed.

Abstract

Please define SOC already in L27 – you wait with a more precise information until L32 (Surgery and doxorubicin) in the abstract. Please define also in the abstract that the control group is a retrospective cohort and not a second prospective study arm.

AUTHORS: done.

Introduction

L72 – please reword “present-days studies”

AUTHORS: done. The paragraph has been rephrased as follows: “Recently, immunotherapy has emerged as a valid combinatorial approach for various types of human cancer. Soft tissue sarcomas have traditionally been considered as immunologically “cold” tumors. However, current studies are re-evaluating the immunotherapy potential in this field.”

L74 and L76: citation are not in square brackets?

AUTHORS: thank you for catching this up.

Material and methods

L100 (p 3): please define “staged negative” or reword

AUTHORS: this has been changed as follows: “To be eligible for recruitment dogs had to be metastasis-free at the beginning of chemotherapy”.

L169 (p 4): Please add that imiquimod was administered at the site of the vaccination, this information is missing here and only mentioned in the discussion. Please give the volume that was administered intradermally.

AUTHORS: this has been changed as follows: “Before each vaccination, imiquimod cream (Aldara 5% cream- a TLR7/8-agonist) was rubbed topically at the site of the vaccination on a 3 x 3 cm shaved and clean area as an additional adjuvant. After 15 minutes, the vaccine (1 ml in total volume) was administered intradermally in close proximity to a peripheral lymph node.”

L 186 (p 4): Please replace “IN” by “In”

AUTHORS: thank you for catching this up.

Results

Figure 1: It seems the legend to 1D is missing?

AUTHORS: thank you for catching this up. The legend has been changed accordingly.

L280 (p 7): the “end of the vaccination schedule” was then after 5 vaccinations or after the real end? (which in 13 dogs was later after 6-11 vaccinations)

AUTHORS: very good comment, thank you. This has been clarified as follows: “Vaccine-specific T-cell and humoral responses were obtained after the fifth vaccination”.

Figure 2: Please explain in 2A T0, T1, etc (L288) as you do it for 2B (L293)

AUTHORS: thank you, this has been added.

L295 Control group: I would consider to present the vaccinated group first and afterwards present the control group, as the control was not a second treatment arm of this study, but a retrospective cohort

AUTHORS: the vaccinated group (demographics and treatment) has been described first (paragraphs 3.2-3.4). Subsequently, the control group has been described (paragraph 3.5). The paragraph on oncologic outcomes follows the previous ones, as it compares the results obtained. We believe that the flow of information is more logical as it is.

Table 1: I would consider to remove the TTP and ST information in the table, as this information is at the moment repeated three times in the manuscript, in Table 1, in the main text and as Figure 3-5.

AUTHORS: TTP and ST have been removed from Table 1, as requested.

Fig 3-5: please reword “stratified”, as these is misleading; there were no two treatment arms and no true stratification

AUTHORS: “Stratified” has been replaced with grouped.

Discussion

L394: Please be consistent with abbreviation of survial time, I see ST, OS and TSS at the moment

AUTHORS: thank you for catching this up.

Reviewer 2 Report

Dear Authors,

This is an exciting study demonstration some benefit from a peptide based vaccination strategy for canine hemangiosarcoma, HSA, a very aggressive disease in dogs and an excellent parallel disease to angiosarcoma in humans. The authors demonstrate significant benefit with regard to time to progression and overall survival in both splenic and cardiac HSA patients. 

The manuscript can benefit from a more clear description of how the vaccine was prepared and administered. Why were only the first and second dose of the vaccine was dissolved in Nobivac Lepto? Which dog received the autologous vaccine? What was the outcome?  Why was the autologous vaccine approach abandoned (as it appears to be used only once). 

In Figure 2b, there appear to be far fewer cases represented than in Figure 2a. Were not all cases sampled?  Please also clarify what T last indicates, as some patients received many more vaccines than others. For the T-cell response in section 2.7, please include how the PBMCs were stored, and how many were plated.

How are the dogs with different numbers of vaccines reflected in the figures relating to the Oncologic outcome, Fig. 3, 4 and 5. 

Please comment further on the control cases. Were all cases presenting to the two institutions with HSA and treated with SOC included for comparison? 

Table 1 lists 1 case in Vaccinated and 1 case in the Non-Vaccinated Dogs as having a treatment related toxicity of grade > 3. Please provide more information regarding these cases and the observation.

There are a few instances where minor revisions can be helpful. For example, line 176  and 178: patients-derived  .. should be... patient derived 

Author Response

Dear Authors,

This is an exciting study demonstration some benefit from a peptide based vaccination strategy for canine hemangiosarcoma, HSA, a very aggressive disease in dogs and an excellent parallel disease to angiosarcoma in humans. The authors demonstrate significant benefit with regard to time to progression and overall survival in both splenic and cardiac HSA patients. 

The manuscript can benefit from a more clear description of how the vaccine was prepared and administered. Why were only the first and second dose of the vaccine was dissolved in Nobivac Lepto?

AUTHORS: The vaccine preparation has been described in two previous publications (23, 24) and briefly reported in the MM paragraph of the current manuscript. Readers can refer to these previous publications for more detailed information. Due to the word count limit, we are not sure whether it is possible to add the whole preparation protocol here (maybe the Editor can comment on this).

Leptospira components are known to effectively activate a vigorous type 1 immune response in mammals (PMID: 12379692, 14505928); we administered Lepto Nobivac in combination with the first and second doses of vaccine for adjuvant purpose. Dogs are routinely vaccinated against Leptospira and we exploited the induced immune response at the site of injection to enhance the efficacy of our vaccine. We applied this strategy to prime the immune response against antigens of the vaccine. For the following four doses were instead relied on the topical imiquimod adjuvant (Aldara), a Toll-like receptor 7/8 agonist that alone was enough to strengthen the pre-existent immune response. We chose Imiquimod also because is often used in human clinical trials (PMID: 30524908). This has been described in the Discussion paragraph.

Naiman BM, Blumerman S, Alt D, Bolin CA, Brown R, Zuerner R, Baldwin CL. Evaluation of type 1 immune response in naïve and vaccinated animals following challenge with Leptospira borgpetersenii serovar Hardjo: involvement of WC1(+) gammadelta and CD4 T cells. Infect Immun. 2002 Nov;70(11):6147-57. doi: 10.1128/IAI.70.11.6147-6157.2002. PMID: 12379692; PMCID: PMC130359.

Brown RA, Blumerman S, Gay C, Bolin C, Duby R, Baldwin CL. Comparison of three different leptospiral vaccines for induction of a type 1 immune response to Leptospira borgpetersenii serovar Hardjo. Vaccine. 2003 Oct 1;21(27-30):4448-58. doi: 10.1016/s0264-410x(03)00439-0. PMID: 14505928.

Trial Watch: Toll-like receptor agonists in cancer immunotherapy. Melody Smith, Elena García-Martínez, Michael R Pitter, Jitka Fucikova, Radek Spisek, Laurence Zitvogel, Guido Kroemer, Lorenzo Galluzzi. Oncoimmunology. 2018 Oct 11;7(12):e1526250.

Which dog received the autologous vaccine? What was the outcome?  Why was the autologous vaccine approach abandoned (as it appears to be used only once). 

AUTHORS: Thank you for this comment. The dog that received the autologous vaccine had a primary axial hemangiosarcoma (ischium). The dog experienced a local recurrence two years after diagnosis, with no evidence of distant metastasis, and the time to metastasis (TTM) was 1215 days. Notably, it was the first dog from which the hemangiosarcoma tumor line was isolated. This has been added (paragraph 2.5). Throughout the study, we used heterologous vaccines as they share the same immunogenic peptides as the autologous ones. These heterologous vaccines are ready-to-use, less expensive, and less time-consuming.

In Figure 2b, there appear to be far fewer cases represented than in Figure 2a. Were not all cases sampled?  Please also clarify what T last indicates, as some patients received many more vaccines than others. For the T-cell response in section 2.7, please include how the PBMCs were stored, and how many were plated.

AUTHORS: Thank you for your comment. Figure 2a reports the assessment of tumor-specific IgG from sera. This type of analysis was performed on all dogs, because all the corresponding sera were available. Conversely, Figure 2b shows T-cell specific activation upon stimulation with HSA-vaccine. In this analysis, we only included patients for whom we had PBMCs in good health (able to proliferate upon stimulation). In some cases, the PBMCs were not sufficiently viable, leading us to exclude those patients from the analysis. The following has been added (paragraph 3.4): “The humoral response was evaluated in all dogs by comparing tumor-specific IgG levels at baseline and after the fifth vaccination (Figure 2A). On the other hand, T-cell specific activation upon stimulation with the vaccine was analyzed in 18 dogs with viable PBMCs after the fifth vaccination (Figure 2B). However, in 10 cases, PBMCs were not sufficiently viable, leading to the exclusion of those patients from the analysis.” This has also been addressed as a limitation.

T last refers to the latest timepoint during the vaccine schedule (MAX T3 = after 5 doses).

The information on how PBMS were stored and how many were plated were provided (paragraph 2.7).

How are the dogs with different numbers of vaccines reflected in the figures relating to the Oncologic outcome, Fig. 3, 4 and 5. 

AUTHORS: We chose not to evaluate the outcome of dogs receiving more than the planned 5 vaccinations due to the fact that, as previously explained, only the dogs that remained metastasis-free after the initial 5 doses were eligible for additional vaccination. This selection process introduces a potential bias in the analysis, leading us to exclude this evaluation from our study.

Please comment further on the control cases. Were all cases presenting to the two institutions with HSA and treated with SOC included for comparison? 

AUTHORS: thank you for this comment. This is correct. We specified this in the MM paragraph: “Dogs treated at the University of Bologna before the start of the trial, that underwent SOC, consisting of surgery and 4 to 6 cycles of adjuvant doxorubicin at the same dosage (i.e., 30 mg/m2 IV every 3 weeks), and that were metastasis-free at the beginning of chemotherapy served as retrospective study cohort controls [27].”

Table 1 lists 1 case in Vaccinated and 1 case in the Non-Vaccinated Dogs as having a treatment related toxicity of grade > 3. Please provide more information regarding these cases and the observation.

AUTHORS: Treatment-related toxicity was in all cases chemo-related. Among vaccinated dogs, we recorded 1 episode of grade 3 neutropenia, while among non-vaccinated dogs, we experienced one episode of grade 3 gastrointestinal toxicity. Adverse events for vaccinated dogs are detailed in the results. Additionally, we have now included chemotherapy-related adverse events in the control group.

Comments on the Quality of English Language

There are a few instances where minor revisions can be helpful. For example, line 176  and 178: patients-derived  .. should be... patient derived 

AUTHORS: thank you. The entire manuscript has been reviewed.